# Nano-spectroscopy of excitons in atomically thin transition metal dichalcogenides

Shuai Zhang[1], Baichang Li[2], Xinzhong Chen[3,4], Francesco L. Ruta[1,5], Yinming Shao[1], Aaron J. Sternbach[1], A. S. McLeod[1], Zhiyuan Sun[1], Lin Xiong[1], S. L. Moore[1], Xinyi Xu[2], Wenjing Wu[6], Sara Shabani[1], Lin Zhou[6], Zhiying Wang[2], Fabian Mooshammer[1], Essance Ray[7], Nathan Wilson[7], P. J. Schuck[2], C. R. Dean[1], A. N. Pasupathy[1], Michal Lipson[8], Xiaodong Xu[7], Xiaoyang Zhu[6], A. J. Millis[1], Mengkun Liu[3,4], James C. Hone[2] & D. N. Basov[1✉]

Excitons play a dominant role in the optoelectronic properties of atomically thin van der Waals (vdW) semiconductors. These excitons are amenable to on-demand engineering with diverse control knobs, including dielectric screening, interlayer hybridization, and moiré potentials. However, external stimuli frequently yield heterogeneous excitonic responses at the nano- and meso-scales, making their spatial characterization with conventional diffraction-limited optics a formidable task. Here, we use a scattering-type scanning near-field optical microscope (s-SNOM) to acquire exciton spectra in atomically thin transition metal dichalcogenide microcrystals with previously unattainable 20 nm resolution. Our nano-optical data revealed material- and stacking-dependent exciton spectra of $MoSe_2$, $WSe_2$, and their heterostructures. Furthermore, we extracted the complex dielectric function of these prototypical vdW semiconductors. s-SNOM hyperspectral images uncovered how the dielectric screening modifies excitons at length scales as short as few nanometers. This work paves the way towards understanding and manipulation of excitons in atomically thin layers at the nanoscale.

---

[1] Department of Physics, Columbia University, New York, NY 10027, USA. [2] Department of Mechanical Engineering, Columbia University, New York, NY 10027, USA. [3] National Synchrotron Light Source II, Brookhaven National Laboratory, Upton, NY 11973, USA. [4] Department of Physics and Astronomy, Stony Brook University, Stony Brook, NY 11794, USA. [5] Department of Applied Physics and Applied Mathematics, Columbia University, New York, NY 10027, USA. [6] Department of Chemistry, Columbia University, New York, NY 10027, USA. [7] Department of Physics, University of Washington, Seattle, WA 98195, USA. [8] Department of Electrical Engineering, Columbia University, New York, NY 10027, USA. ✉email: db3056@columbia.edu

Excitons, bound electron–hole pairs, govern the optical properties of two-dimensional transition metal dichalcogenides (TMDs) over near-infrared (IR) and visible frequencies due to their large binding energy and prominent oscillator strength[1]. Notably, all excitonic parameters in TMDs, including their resonance energies as well as both radiative and non-radiative lifetimes, can be manipulated on demand. Relevant control mechanisms include electric and magnetic fields, heterostructuring, moiré superlattice potentials[2–5], local strain[6], ultrafast optical excitation[7,8], dielectric screening[9], among others. The vast majority of experimental studies of excitonic phenomena in TMDs have been carried out by far-field optical methods with wavelength-limited spatial resolution. Nanoscale probes of the engineered excitonic electrodynamics are required to explore heterogeneous features. Nano-probe experiments reported here have the potential to map out the spatial evolution of exciton resonance energies in structures with (spatially) controlled dielectric screening and interlayer hybridization. In addition, nano-spectroscopy can document the potential role of extrinsic factors, such as defects and impurities, especially wrinkles and contaminations common in atomically layered heterostructures. Finally, applications of TMDs in nanophotonics and optoelectronics will benefit from characterization of excitons in nanodevices with sub-diffractional feature sizes that we demonstrated in our study.

A variety of scanning probe optical methods are in general suitable for the nanoscale characterization of the excitonic properties. Specifically, tip-enhanced photoluminescence (TEPL) studies have uncovered aspects of nanoscale excited state recombination in TMDs[10–19]. Complementary to the TEPL, absorption spectra encode not only the exciton resonance energy but also the oscillator strength and damping rate, offering additional detailed inquiry into the excitonic response. In principle, local absorption spectra acquired by scattering-type scanning near-field optical microscopy (s-SNOM) allow one to obtain the complex dielectric function[20], $\varepsilon(\omega) = \varepsilon_1(\omega) + i\varepsilon_2(\omega)$ and make it possible to probe the engineered excitons at the nanoscale. This latter capability has been extensively used to study plasmons and phonons in other classes of van der Waals (vdW) materials[21–23]. However, formidable challenges so far precluded quantitative nano-spectroscopy of excitonic absorption at near-IR/visible (NIR/Vis) by means of s-SNOM[24]. For example, the reduced wavelength gives rise to more scattered light from the AFM tip shank, resulting in increased artificial background; the s-SNOM phase is more prone to optical path length drift as the wavelength is reduced.

Here, we coupled an ultra-stable and tunable continuous-wave Ti-sapphire laser (M squared) to a commercial s-SNOM (Neaspec GmbH) and investigated the nanoscale excitonic spectra in monolayer TMD semiconductors and heterostructures. The materials of interest are monolayers of $WSe_2$ and $MoSe_2$, $MoSe_2/WSe_2$ heterobilayers, as well as $WSe_2$ trilayers. The exciton resonance energy, radiative lifetime, and the damping rate in vdW monolayers were extracted from the s-SNOM spectra with hitherto unattainable spatial resolution of 20 nm. After documenting the ability of s-SNOM to probe excitons in atomically thin semiconductors with the nanoscale spatial resolution, we investigated heterostructures with lateral dimensions smaller than the diffraction limit of NIR/vis light. We show that intralayer excitons in $MoSe_2/WSe_2$ heterostructures red-shift relative to their counterparts in isolated monolayers due to the enhanced dielectric screening. The screening length is found to be shorter than the spatial resolution (20 nm) of our experiments and is attributed to the small radius of tightly bound excitons. The s-SNOM spectra were also applied to probe excitons altered by interlayer hybridization in a multi-layer $WSe_2$ crystal.

## Results and discussion

**Experimental methods and samples preparation.** The s-SNOM is based on a tapping mode atomic force microscope (AFM). A Pt–Ir coated AFM tip is illuminated by a focused NIR/vis beam from a tunable continuous-wave laser, as shown in Fig. 1a. An intense optical field forms in the vicinity of the tip apex and interacts with the TMD crystals underneath it. The light backscattered from the tip is registered by a silicon detector and then demodulated at the high harmonics of the tip-tapping frequency via the pseudo-heterodyne interferometric scheme. This demodulation method allows one to isolate the genuine near-field signals from the undesired far-field background[25,26]. By scanning the tip on the sample, complex near-field signal is recorded with a resolution of ~20 nm (Fig. 1b and Supplementary Fig. 1) and denoted as $se^{i\varphi}$, where $s$ and $\varphi$ are amplitude and phase, respectively. Atomically thin TMD monolayers and their heterostructures were prepared by mechanical exfoliation and stamped on a h-BN substrate. The spectral range used to investigate the samples is 1.50–1.75 eV, which covers the 1s exciton resonance energies of $WSe_2$ and $MoSe_2$ monolayers (see inset in Fig. 1b). In this energy range, the near-field signal from h-BN is frequency-independent[27] and can be utilized as reference for spectroscopic data for TMD materials. The normalized near-field amplitude and phase at the $n$-th harmonic and energies $\omega$ are denoted by $s_n(\omega)/s_n(h-BN)$ and $\varphi_n(\omega) - \varphi_n(h-BN)$, respectively. A combination of our ultra-stable laser and scanner system allowed us to demodulate the scattering signal at the fifth harmonic of the tapping amplitude; high harmonic data is imperative for obtaining genuine near-field contrast devoid of far-field artifacts.

To display the excitonic responses in the near field, in Fig. 1c–e we show representative s-SNOM images for a sample with monolayers of $MoSe_2$ and $WSe_2$, $MoSe_2/WSe_2$ heterobilayer, and unobscured h-BN all in the same field of view (FOV) of our apparatus. An AFM topographic image is shown in Fig. 1c, where a substantial height variation from left to right is prompted by a terrace in the underlying h-BN substrate. Nevertheless, regions of monolayer, bilayer, and multilayer TMDs can be readily identified by their topographic height. Our layer thickness assignment is further confirmed by the second harmonic generation (SHG) data (see Supplementary Figs. 2 and 3). The co-located images of topography (Fig. 1c), near-field scattering amplitude (Fig. 1d), and phase (Fig. 1e) are all acquired simultaneously. The amplitude image (Fig. 1d) reveals nearly uniform signals collected within terraces demarcated by boundaries between the terraces. We acquired images in Fig. 1d, e by setting the excitation photon energy at 1.68 eV. At this energy, a strong phase contrast between the $WSe_2$ monolayer and h-BN substrate is observed (Fig. 1e). On the contrary, no discriminable phase contrast is found for the $MoSe_2$ monolayer. The observation is readily understood as the near-field phase is related to the absorption[28]. The different excitonic absorption characteristics of the TMDs can give rise to phase contrast. According to our hyperspectral data in Fig. 2, the photon energy of 1.68 eV is close to the exciton resonance in $WSe_2$ monolayer and does not overlap with the excitonic band of $MoSe_2$ centered at 1.58 eV. Our data reveal that the prominent nano-optical contrast in Fig. 1 is rooted in the local excitonic response of TMD monolayers. It should be noted that the contributions of thermal effect and charge effect to the near-field contrast can be ruled out by the high harmonic demodulation as also evident in the near-field spectra shown in following sections.

## Near-field nano-spectroscopy of excitons in $WSe_2$ monolayer.

For quantitative analysis of the $WSe_2$ monolayer excitonic response, we carried out raster-scanned nano-imaging while

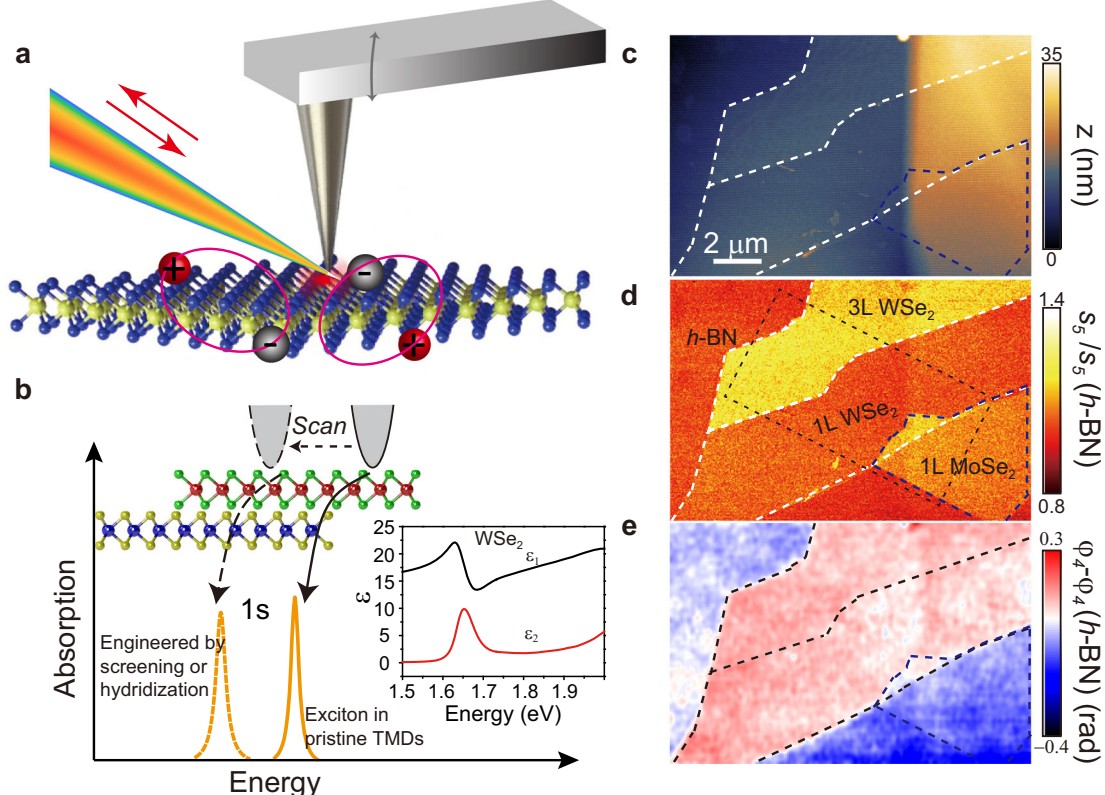

**Fig. 1 Near-field nano-probing of excitons in atomically thin transition metal dichalcogendies (TMDs). a** Schematic illustration of near-field measurement of excitons. The metallized AFM tip is illuminated by focused incident light and the tip-scattered light is collected. Pairs of electrons and holes with strong Coulomb interactions in atomically thin TMDs form excitons with large binding energies. The nanoscale response of excitons is extracted from the back-scattered light. **b** Scanning the tip across the TMD sample allows one to investigate the excitonic response below the diffraction limit and extract the dielectric function at the nanometer scale. The inset shows the dielectric function of monolayer WSe$_2$ from ref. [61] **c** An AFM topographic image of the TMDs in this study. The height difference between the left and right regions is due to the h-BN terrace. **d, e** Near-field images of normalized scattering amplitude $s_5$ and phase $\varphi_4$ on the same region of (**c**). The excitation energy is 1.68 eV. The dashed bright and blue lines trace the edges of the different sample regions. The dashed rectangle marks the position of Fig. 2a.

varying the photon energy with a step size of ~10 meV. We have simultaneously collected topographic and near-field maps at each energy over the same FOV marked by the dashed rectangle in Fig. 1d. The topography image and height profile are shown in Supplementary Fig. 3. The unobscured h-BN in the upper left corner of Fig. 2 is used as a reference for normalization purposes. Representative maps of the normalized fifth-harmonic amplitude $s_5(\omega)$ and phase $\varphi_5(\omega)$ are shown in Fig. 2a, b, respectively. In these images, both the amplitude and phase signals in WSe$_2$ monolayers vary systematically as a function of excitation energy. In particular, the phase images display a prominent contrast centered around 1.66 eV. Notably, we witness a highly uniform response within regions occupied by each of the TMD materials in Fig. 2. All the boundaries and some terraces are in a deeply sub-diffraction regime. Nevertheless, these features can be clearly discriminated in our s-SNOM images. To quantify the excitonic response, we start our analysis by averaging the near-field signals over the entire region occupied by WSe$_2$ monolayer at each probing frequency. The averaging process will facilitate a more faithful determination of dielectrics function, which will be discussed later. The net result is that each data point in spectra plotted in Fig. 2c, d is representative of the entire region imaged at the corresponding energy in Fig. 2a, b. The spectra derived from images show prominent resonance characteristics: the amplitude signal reveals a dispersive behavior around 1.66 eV, whereas the phase signal exhibits a peak centered at the same energy.

We modeled experimental near-field spectra of WSe$_2$ in Fig. 2 with the Lorentzian form of the dielectric function[29–31]:

$$\varepsilon(\omega) = \varepsilon_\infty - \frac{c}{\omega_0 d}\frac{\gamma_{r,0}}{\omega - \omega_0 + i\left(\frac{\gamma_{nr}}{2} + \gamma_d\right)} \qquad (1)$$

where $\varepsilon_\infty$ is the high-frequency permittivity limit and originates from all transitions with frequencies beyond the spectral range investigated here, $c$ is the speed of light in vacuum, $\omega_0$ is the exciton resonance energy, $d$ is the monolayer thickness, $\gamma_{r,0}$, $\gamma_{nr}$, $\gamma_d$ are radiative, non-radiative, and dephasing decay rates, respectively. The oscillator strength of the Lorentzian in Eq. 1 is related to the radiative rate $\gamma_{r,0}$. We choose the point dipole model[32] to interpret the data as it is well documented to capture the response of atomically thin samples laid on thick substrates[32,33]. As shown in Fig. 2c, d, the point dipole model with Eq. 1 as the input dielectric function provides an adequate fit to the data (see SOM for details on fitting).

Now we analyze the parameters inferred from the fitting procedure (Table 1). The obtained dielectric function is plotted in Fig. 2e. The radiative rate for monolayer WSe$_2$ is 1.4 meV, which corresponds to the radiative lifetime $\tau_0 = \frac{\hbar}{2\gamma_{r,0}}$ of ~233 fs. This latter finding is consistent with the first principles theoretical values[34] and experimental results derived from the ultrafast exciton 1s-2p transition, which reveal 150 fs radiative lifetime in WSe$_2$ monolayers[35]. The shorter radiative lifetime compared to that of conventional semiconductors stems from the exceptionally

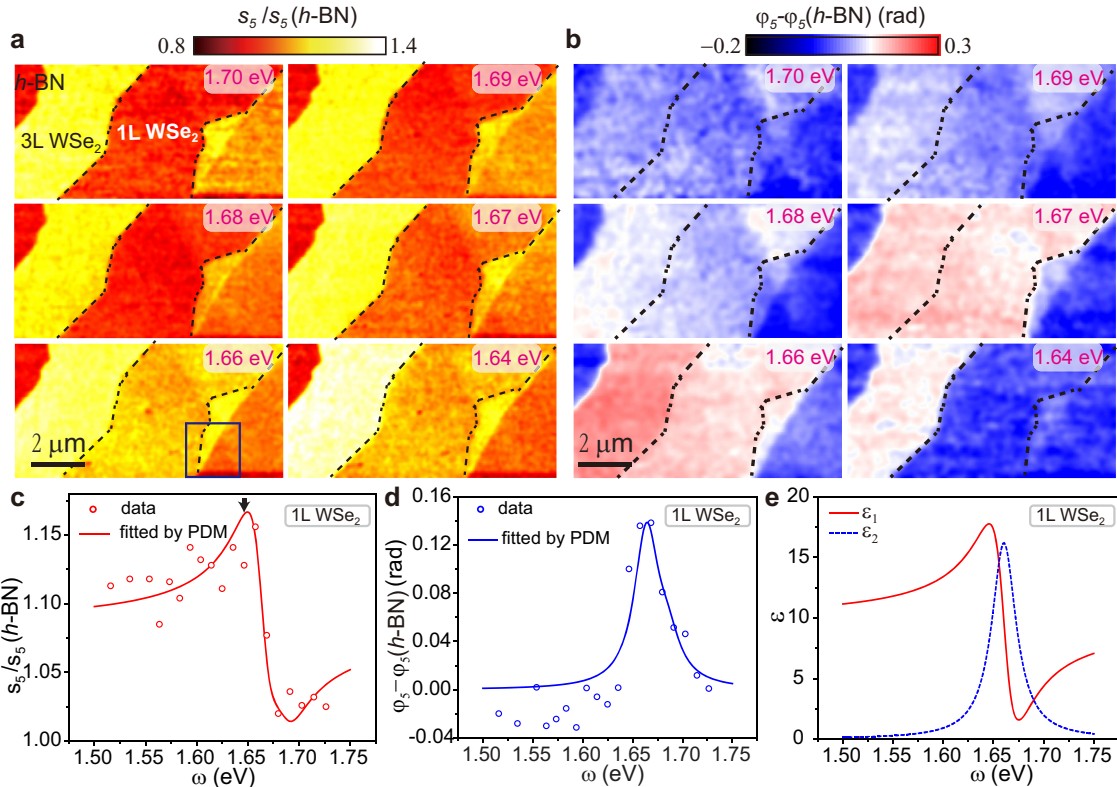

**Fig. 2 Evolution of the scattering amplitude and phase for the monolayer WSe₂ as a function of the excitation energy. a** Representative near-field images of the normalized scattering amplitude $s_5(\omega)/s_5(h-\mathrm{BN})$. The excitation energies are indicated in the images, which are acquired by scanning over a rectangular area marked with the dashed line in Fig. 1d. Boundaries of the monolayer WSe₂ region are displayed with black dashed lines. The area enclosed by blue lines in bottom left image was investigated in Fig. 4a. **b** Near-field images of the normalized phase $\varphi_5(\omega) - \varphi_5(h-\mathrm{BN})$. The images with the same excitation energy in **a** and **b** are acquired simultaneously. An analogous dataset demodulated at the fourth harmonic is shown in Supplementary Fig. 5. **c, d** Normalized near-field amplitude $s_5(\omega)/s_5(h-\mathrm{BN})$ and phase $\varphi_5(\omega) - \varphi_5(h-\mathrm{BN})$ spectra for WSe₂ monolayer (data points). Fits to the spectra using the point dipole model (PDM) and a single Lorentz oscillator (Eq. 1) are shown as solid curves. The arrow in **c** marks the peak energy of PL spectra (Supplementary Fig. 4). **e** Dielectric function of WSe₂ monolayer extracted from the fits with PDM in (**c, d**).

**Table 1 The parameters of Lorentz model used to describe TMD dielectric function as obtained from fitting s-SNOM spectra with the point dipole model.**

|  | 1L WSe₂ | 1L MoSe₂ | MoSe₂/WSe₂ bilayer | 3L WSe₂ |
|---|---|---|---|---|
| High-frequency permittivity limit $\varepsilon_\infty$ | 9.7 ± 0.5 | 15.6 + 0.7 | 13.6 ± 0.7 | 13.8 ± 0.8 |
| Exciton energy $\omega_0$ (meV) | 1660 ± 2 | 1583 ± 3 | 1573 ± 5 (MoSe₂) | 1639 ± 3 |
|  |  |  | 1648 ± 6 (WSe₂) |  |
| Radiative rate $\gamma_{r,0}$ (meV) | 1.4 ± 0.2 | 1.7 ± 0.3 | 0.9 ± 0.5 (MoSe₂) | 5.4 ± 0.7 |
|  |  |  | 2.1 ± 0.6 (WSe₂) |  |
| $\frac{\gamma_{nr}}{2} + \gamma_d$ (meV) | 15 ± 3 | 18 ± 4 | 13 ± 8 (MoSe₂) | 20 ± 4 |
|  |  |  | 24 ± 9 (WSe₂) |  |

large exciton oscillator strength in two-dimensional TMDs, evidenced by the high optical absorption in monolayer[36,37]. We obtained the damping rate $\frac{\gamma_{nr}}{2} + \gamma_d = 15$ meV. We remark that this substantial damping at room temperature inhibits the real dielectric function from crossing zero, and thus the exciton polariton regime cannot be reached. Recently, reflection measurements at cryogenic temperature demonstrated the negative real dielectric function in the vicinity of the excitonic line[38]. Therefore, by reducing the damping rate, for example, through experiments conducted at cryogenic temperature[31,38], it may become possible to access the exciton polaritons in monolayer TMDs. So far, all experiments on exciton polaritons in TMDs

conducted at ambient required integration of the active semiconductor in various forms of photonic cavities[39–41]. A notable exception is an observation of the excited state Rydberg exciton polaritons demonstrated in pump-probe experiments for multilayer WSe₂ at room temperature[8].

Next, we comment on the contributions of bright and dark excitons to the s-SNOM spectra. These two types of excitonic species result from the spin splitting of the conduction band edge and they obey different optical selection rules[42,43]. In TMD literature, excitons that can be excited by light with polarization in the sample plane, as in normal incidence experiments, are referred to as bright excitons. Excitons that can be activated only

when light polarization is perpendicular to the sample plane, are referred to as dark excitons. Dark excitons can be visualized using TEPL for TMD samples on metallic substrates because the electric field strength perpendicular to the sample surface is drastically enhanced[12,19]. In contrast, the s-SNOM signal for atomically thin TMDs is governed by the in-plane dielectric function[32,44], which arises from bright excitons (see SOM). To experimentally confirm the dominant role of bright excitons in our data, we compared the bright exciton resonance energy obtained from far-field PL spectra on the same sample[45] with the resonance energy in our s-SNOM spectra. The bright exciton resonance energy from PL (see Supplementary Fig. 4) is about 1.66 eV, in close agreement with s-SNOM data. We, therefore, conclude that bright excitons dominate the s-SNOM response of monolayer and few-layer TMD crystals.

As discussed above, the s-SNOM spectra are determined by the dielectric function of the materials; whereas TEPL is determined by both the exciton population and the transition rate[46] and therefore is sensitive to sample parameters. Stain, defects, and contaminants in samples, particularly chemical vapor deposition (CVD) grown ones commonly used in TEPL experiments, can impact the exciton distribution and resonance energies and thus give raise to inhomogeneity of the TEPL. Here, to probe the intrinsic properties of TMDs, the samples are exfoliated from the high quality crystals grown by the self-flux method[47] and fabricated by the contamination-free stacking method (details in "Methods"). The resulting high-quality sample possesses nearly uniform dielectric function, thereby exhibiting a nearly homogenous s-SNOM signal.

**Excitonic spectra of TMD monolayers**. Before exploring the nanoscale response of heterostructures and multilayers, we take a closer look at the near-field response of the constituent monolayers. The spectra of the normalized amplitude and phase for MoSe$_2$ are shown in the middle panels of Fig. 3a, b (see Supplementary Fig. 6 for complete data sets). Each data point in these spectra is attained by averaging the near-field signal over the entire MoSe$_2$ regions. The near-field spectra for MoSe$_2$ are akin to those of WSe$_2$ and exhibit nearly the same resonance features, manifesting as a derivative-type lineform of the amplitude and a peak in the phase. The resonance energy observed in MoSe$_2$ using s-SNOM (1.583 eV) is consistent with the dominant peak in PL spectra[48] (see Supplementary Fig. 4), supporting the notion of bright excitons. The radiative rate obtained by fitting the spectra for MoSe$_2$ is as high as 1.7 meV and is slightly larger than that of WSe$_2$ (see Table 1). In general, the prominent radiative rates stem from the two dimensional nature of excitons in TMDs, which are tightly bound to monolayers and have a small exciton radius[49]. Even though the spectral lineforms of excitonic lines in MoSe$_2$ and WSe$_2$ are nearly identical, the near-field amplitude detected for MoSe$_2$ is considerably larger than that of WSe$_2$ below the exciton resonance energies (top and middle panels of Fig. 3a). We also found that the magnitude of the real part of the dielectric function $\varepsilon_1$ at energies below the exciton resonance is enhanced in MoSe$_2$ compared to that of WSe$_2$. This latter finding is consistent with the enhanced radiative rate in MoSe$_2$.

**MoSe$_2$/WSe$_2$ heterobilayers and WSe$_2$ trilayers**. Now we turn to the spectra acquired for the MoSe$_2$/WSe$_2$ heterostructures with the width of less than 1 μm, as shown in the $s_5$ amplitude map (Fig. 3c, where HB denotes the heterobilayer region). The amplitude and phase spectra are displayed in the bottom panels in Fig. 3a, b, respectively. In these spectra, two resonances emerge at the energies close to the frequency positions of the exitonic

bands in MoSe$_2$ and WSe$_2$ monolayers in Fig. 2 and Supplementary Fig. 6, indicating a dominant contribution of intralayer bright excitons. The role of interlayer excitons remains negligible in the data plotted in Fig. 3. Indeed, the resonance energies of indirect excitons[4,50,51] (~1.3–1.4 eV) fall outside the spectral range investigated in Fig. 3.

We analyzed spectra of the MoSe$_2$/WSe$_2$ heterostructure assuming that the dielectric function is given by the sum of two Lorentzians in Eq. 1. The fit with PDM shows that both the intralayer exciton resonance energies red-shift compared to their counterparts in monolayers (vertical dashed lines in Fig. 3a, b) Red-shift of the exciton resonance energies can also be clearly identified in the hyperspectral measurements of the amplitude $s_5$ and phase $\varphi_5$ (in Fig. 3d, e, respectively) along the line trace in Fig. 3c. The frequency red-shift of the two exciton resonances is around 10 meV compared to their monolayer counterparts (Table 1 and Fig. 3a, b).

To unravel the origin of the red-shifts, the twist angle of the two layers comprising our heterostructure was characterized by polarization-resolved SHG experiments. The SHG data show that the twist angle between layers is either 15° or alternatively 45° (see Supplementary Fig. 2). Either of the two layer arrangements implies the misalignment between the Brillouin zones of monolayers[51,52] and therefore produces only negligible hybridization between the two atomic planes. However, excitons in the heterostructure experience stronger dielectric screening compared to the isolated monolayers, because the dielectric function of TMDs is larger than that of h-BN (~4). Therefore, we conclude that in heterostructures with a large twist angle, the redshift of intralayer exciton resonance energies mainly arises from dielectric self-screening[9,52].

Although the effect of dielectric screening on excitons has been widely investigated, the screening length is difficult to infer from previous reports based on diffraction-limited far-field data[9,53]. Here the screening length refers to the spatial span over which the exciton energies are altered by abrupt changes in the dielectric environment. s-SNOM experiments visualize the spatial evolution of the exciton lines across interfaces and thus can potentially quantify the screening length. Representative s-SNOM images in Fig. 4a, b show that the amplitude and phase contrasts abruptly change across the boundary between the monolayers and heterostructure. From the high-resolution images and the line profiles (Fig. 4c), one can see that the exciton resonance energy is altered over the length scale of about 20 nm, which is the spatial resolution in our experiment. We conclude that our nano-NIR/vis scans yield only the upper bound estimate of this length scale limited by the spatial resolution of our method. Our estimate is consistent with the theoretically predicted value of about 10 nm (ref. [9]). Figure 4a–c displays the representative results affirming the s-SNOM contrast changes on the length scale of ~20 nm. Data in Figs. 2 and 3 uncovers the abruptness of these changes both in amplitude and phase signals over the entire frequency range. The totality of the data in Figs. 2–4 attest to the nanoscale sensitivity of our approach to the dielectric function.

In addition to the dielectric screening, interlayer hybridization also plays a role in controlling the excitonic properties of vdW materials, particularly in homostructures. Our nano-NIR/vis data provides an experimental access to interlayer hybridization in trilayer WSe$_2$ exfoliated from 2H-stacked crystal. We anticipate that in these crystals neighboring layers are twisted by an angle of 60°. The amplitude and phase spectra acquired on the WSe$_2$ trilayer are shown in Fig. 4d, e, respectively, with fitting results presented in Table 1. These data show that the exciton resonance energy decreases from 1.660 eV in WSe$_2$ monolayer (dashed vertical line in Fig. 4e) to 1.639 eV in the trilayer (solid vertical

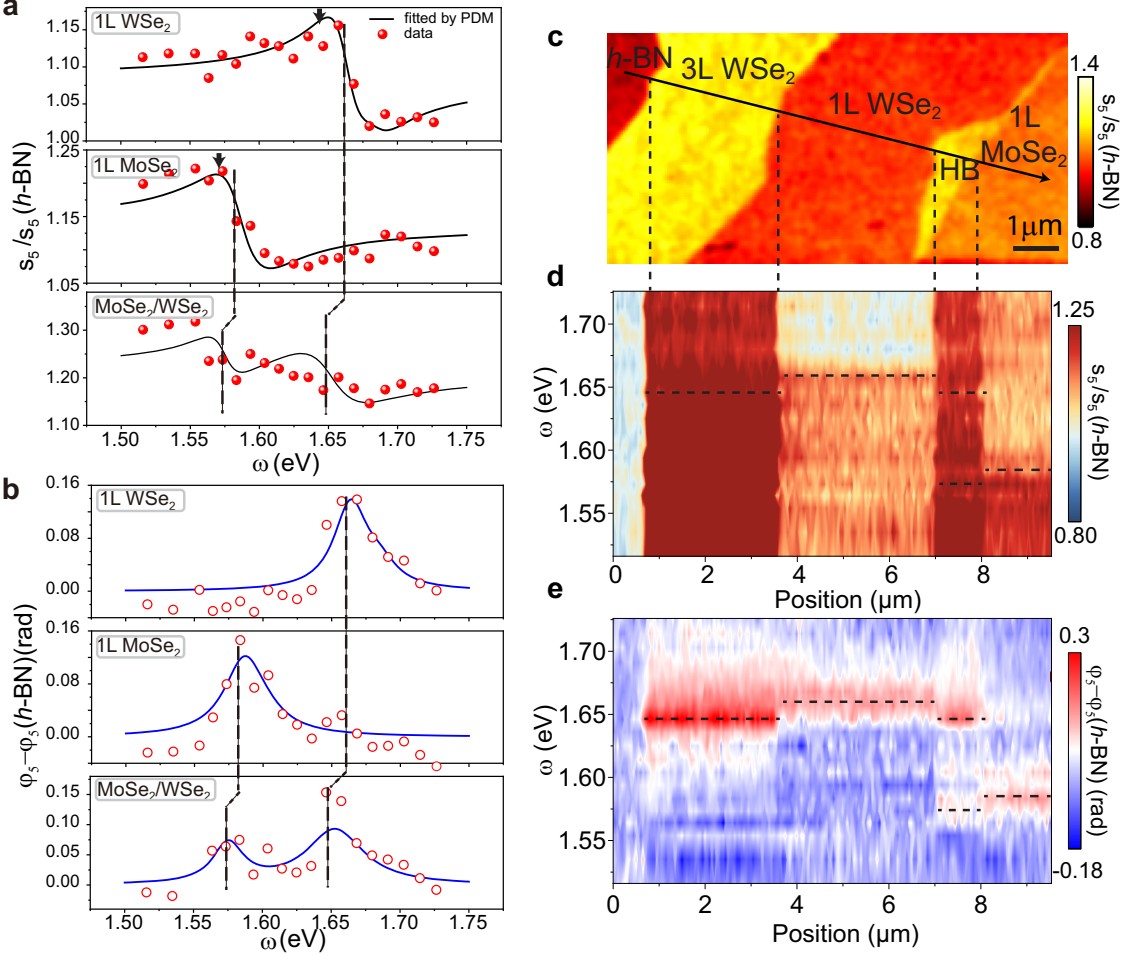

**Fig. 3 Nano-spectroscopy of different types of excitons. a, b** Normalized amplitude $s_5(\omega)/s_5(h-BN)$ and phase $\varphi_5(\omega) - \varphi_5(h-BN)$ spectra for different sample regions (data points). The near-field data is fitted by the point dipole model and Lorentz oscillators (solid curves). The vertical dashed lines are used to mark the exciton energies of $WSe_2$ and $MoSe_2$ extracted from the point dipole model. The arrows in **a** mark peak energies of the PL spectra (Supplementary Fig. 4). **c** Near-field image of the amplitude $s_5$ taken at an excitation energy of 1.52 eV. HB: heterobilayer. **d, e** Near-field amplitude $s_5$ and phase $\varphi_5$ evolution along the line trace shown in (**c**). The horizontal dashed lines in **d** and **e** denote the exciton energies extracted from the point dipole model.

lines in Fig. 4e). The indirect transitions in few-layer TMD crystals have negligible oscillator strength, and thus make only a minute contribution to the dielectric function and/or to the s-SNOM signal. Therefore, the observed redshift of the exciton resonance energy originates from direct transitions. For trilayer $WSe_2$ with a twist angle of 60°, interlayer electronic hybridization splits both conduction and valence band edges at the corner of the Brillouin zone, thus reducing the energy gap and the corresponding exciton resonance energy[52,54,55]. Therefore, the large shift observed in $WSe_2$ trilayer (21 meV) is likely to be caused by the interlayer hybridization. We note that the dielectric screening could also reduce the exciton resonance energy, but its effect is subtle (~10 meV, as demonstrated in the last section), compared to the interlayer hybridization.

In summary, the bright excitonic responses of TMD monolayers were revealed by s-SNOM hyperspectral nano-imaging. New data allowed us to extract the exciton resonance energy, oscillator strength, and damping rate all with the nanometer resolution. Nano-NIR/vis spectra and images uncovered the spatial evolution of the dielectric screening and interlayer hybridization. We obtained the upper bound of the dielectric screening length of 20 nm limited by the achieved spatial

resolution of s-SNOM in this work. Our results lay the groundwork for future spatio-temporal[35,53] studies of excitonic states, including moiré excitons[2,56–58], exciton liquid[59], and exciton phase transitions[60] in a wide range of quantum materials.

## Methods

**Sample preparation.** Monolayers of $WSe_2$, $MoSe_2$, and 30 nm thick BN were exfoliated onto $SiO_2/Si$ using the standard scotch tape method. To prepare the heterostructure sample, a polydimethylsiloxane (PDMS) stamp was coated with thin polypropylene carbonate (PPC) film and subsequently attached onto a glass slide. Inverted stacking was performed on a dry transfer station with a rotatable heating stage. Thick BN and monolayers of $WSe_2$ and $MoSe_2$ were picked up subsequently, with controlled alignment. Finally, the heterostructure with PPC was peeled off from the PDMS and transferred to a clean $SiO_2/Si$ substrate at 120 °C. The twist angle of the heterostructure is determined by the SHG spectra from individual layers and their overlap region.

**Near-field measurement.** The nano-imaging was performed using a commercial s-SNOM (www.neaspec.com) based on a tapping-mode AFM. The tapping frequency and amplitude of the AFM are about 75 kHz and 50 nm, respectively. The light source is a tunable continuous laser from M Squared, including single frequency CW-532 nm module (EQUINOX), continuous-wave Ti-sapphire model with output range 700–1000 nm (SolsTiS), and frequency mixing module with tunable output range 1100–2200 nm (DFG-532). By focusing the laser beam onto the metallized AFM tip, we were able to probe the optical properties with

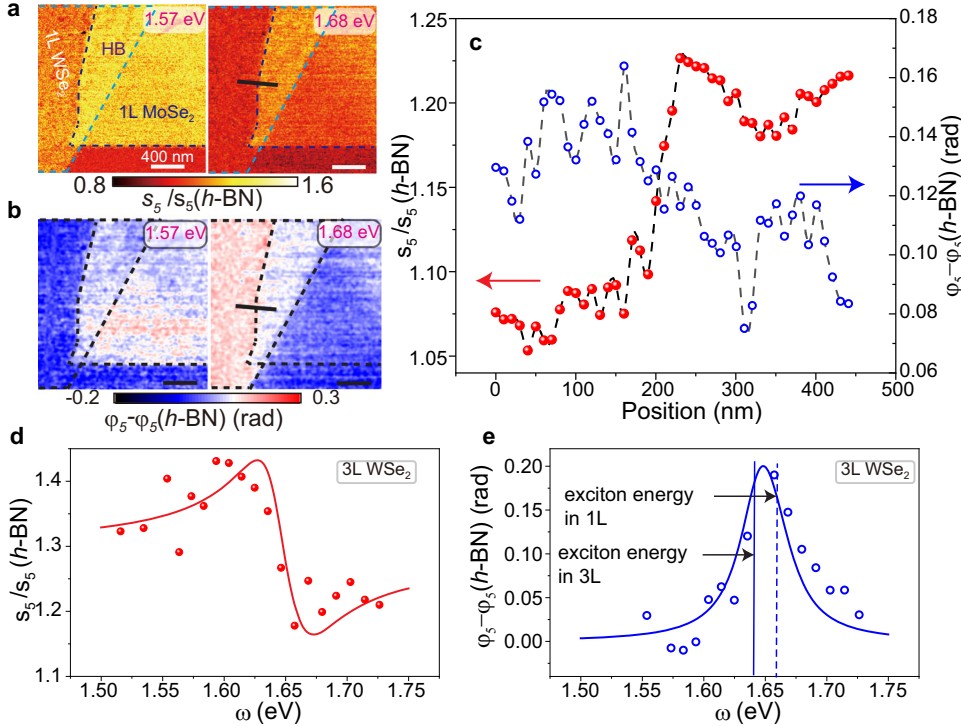

**Fig. 4 Screening length of excitons in heterostructure and the effect of hybridization on the exciton resonance in WSe₂ trilayer. a, b** Near-field images of the normalized amplitude $s_5(\omega)/s_5(h-BN)$ and phase $\varphi_5(\omega)-\varphi_5(h-BN)$. The images in the left and right panels were taken at 1.68 eV (around the monolayer WSe₂ exciton resonance energy) and 1.57 eV (around monolayer MoSe₂ exciton resonance energy), respectively. The boundaries of the materials are displayed with dash lines. The scale bars indicate 400 nm. **c** Line profiles of the normalized amplitude and phase across the monolayer and heterobilayer along the black lines in the right panels in (**a** and **b**). **d, e** Normalized scattering amplitude $s_5(\omega)/s_5(h-BN)$ and phase $\varphi_5(\omega)$ spectra for WSe₂ trilayer (data points). The near-field data are fitted by the point dipole model (solid curves). The vertical blue dashed and solid lines in e are used to mark the exciton energies of WSe₂ monolayer and WSe₂ trilayer extracted from the point dipole model, respectively.

subwavelength resolution. To this end, the back-scattered light is registered by pseudo-heterodyne interferometric detection and then demodulated at the $n$-th harmonics of the tapping frequency yielding background-free images. To eliminate the background, we chose $n = 4$ and 5 in this work. In addition to the background elimination, the high harmonic signal can also reduce the detection depth, which is essential to detect the atomically thin TMDs.

## Data availability

Raw files containing the unprocessed near-field images are available from the corresponding author upon reasonable request.

## Code availability

Code used to fit the near-field spectra is available from the corresponding author upon reasonable request. This code is based on the NearFieldOptics python package, downloaded from a github repository from Alexander S. Mcleod (https://github.com/asmcleod/NearFieldOptics).

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

## Acknowledgements
Experimental research at Columbia is supported as part of Programmable Quantum Materials, an Energy Frontier Research Center funded by the U.S. Department of Energy (DOE), Office of Science, Basic Energy Sciences (BES), under award DESC0019443. F.M. gratefully acknowledges support by the Alexander von Humboldt Foundation. Authors thank Yilei Li for providing the dielectric function data of WSe$_2$.

## Author contributions
S.Z. and D.N.B. designed the experiments. B.C.L., W.J.W., S.S., L.Z., Z.Y.W., E.R., N.W., C.R.D., A.N.P., X.X., X.Y.Z. and J.H. designed and fabricated the devices. X.Y.X. and P.J.S. characterized the sample twist angle. S.Z. performed the experimental measurements with input from Y.M.S., A.J.S. and L.X. S.Z. and D.N.B analyzed the experimental data. X.Z.C., M.K.L. and F.L.R. developed the theoretical model to analyze the experimental data with input from A.J.S., A.S.M., Z.S., S.L.M., F.M., M.L. and A.J.M. S.Z. and D.N.B. co-wrote the manuscript with input from all co-authors. D.N.B. supervised the project. All authors contributed to the discussions.

## Competing interests
The authors declare no competing interests.
