## [Peer Review File · Nature Communications]

Nano-spectroscopy of excitons in atomically thin transition metal dichalcogenidesREVIEWER COMMENTS

Reviewer #1 (Remarks to the Author):

The authors present experimental studies of s-SNOM imaging of excitons in monolayer MoSe₂, WSe₂, heterostructure and trilayer WSe₂ materials. They demonstrate measurements of dielectric functions and claim 20 nm spatial resolution. The results allowed for determining the 20 nm upper bound for dielectric screening length. The work is novel and interesting and may be published after the following comments are addressed:

- 1) On page 6 the authors claim that “data reveals that the prominent nano-optical contrast ... is rooted in the local excitonic response of TMD monolayers.” That is a reasonable expectation. However, is there an alternative explanation based on the tip shadowing effect? For example, the scattering signal at modulated harmonic is sensitive to the behavior of the tip. The tip oscillation changes across the junction between two materials because of different tip-sample interactions even in the absence of incident laser. Nonresonant scattered light will also give contrast. Resonant excitation may give larger contrast because of heating or charging effects. Both of these may give sub-diffraction contrast limited by the size of the tip even without localized excitation. So the claim of the local excitonic contribution is questionable. It would be helpful to clarify the mechanism behind the contrast. For example, the image showing lack of contrast for nonresonant excitation would prove the proposed local excitonic effect. For example, Figure 4a shows the presence of contrast between WSe₂ and hBN with nonresonant excitation. Does that contradict the proposed localized excitonic mechanism?
- 2) Why do the authors use large scale averaging to obtain dielectric function spectra? Is it because of low SNR? Does that limit the application of s-SNOM?
- 3) The authors claim uniform distribution of s-SNOM signal over the flakes. That contradicts the previous findings of TEPL that showed heterogeneous local excitonic response. How do the authors interpret it? In fact, from the s-SNOM images it seems that there are some heterogeneous structures on top of the uniform background. They might even be correlated for different excitations. Did the authors analyze such correlations?
- 4) In Table 1 what is the meaning of epsilon_bg and why is it different for different materials? Is it related to the dielectric function of hBN that should have the same value for any location?
- 5) The claims of nanoscale resolution are questionable in the context of resonance energies and dielectric functions. There is no data in the manuscript that shows either of them with nanoscale resolution. The only data are for the s-SNOM signals at one excitation energy at a time. The authors should either revise the claims in the abstract and summary or add more data. For example, Figure 3e shows a promising result but in microscale. Did the authors try to zoom in or the SNR is too small?

Reviewer #2 (Remarks to the Author):

- What are the noteworthy results?

This is an excellent work where the authors use s-SNOM technique coupled with near-infrared and visible frequency lasers to perform spectroscopy and map the real and imaginary parts of the dielectric functions of ML MoSe₂, WSe₂, and their heterostructures. They implement the method of the equivalence of far-field absorption spectra with the s-SNOM phase spectra to study excitons in these structures. Along the way they also found the effect of dielectric screening on the dielectric function.

- Will the work be of significance to the field and related fields? How does it compare to the established literature? If the work is not original, please provide relevant references.

The work is significant in the sense that nano-spectroscopy in MLs and heterostructures has not been applied before and this now opens a way to study excitons at high-resolution.

- Does the work support the conclusions and claims, or is additional evidence needed?

The work is supported by the conclusions and claims, the reviewer has made some comments to improve the presentation of the work.

- Are there any flaws in the data analysis, interpretation and conclusions? Do these prohibit publication or require revision?

No

- Is the methodology sound? Does the work meet the expected standards in your field?

Yes

- Is there enough detail provided in the methods for the work to be reproduced?

Yes

Comment 1:

In lines 68-70 the authors say:

“ This latter capability

68 has been extensively used to study plasmons and phonons in other classes of van der Waals
69 (vdW) materials²¹⁻²³. However, formidable technical challenges so far precluded
70 quantitative nano-spectroscopy of excitonic absorption by means of s-SNOM.”

I believe what the authors mean by formidable technical challenge is just simply lack of stable lasers in the visible and near-IR frequencies, nothing else. Also note similar visible/near-IR nano-spectroscopy with s-SNOM has been done before, although not on MLs see for example APPLIED PHYSICS LETTERS 96, 063107 (2010) . For all these reasons I would simply say:

“This latter capability

68 has been extensively used to study plasmons and phonons in other classes of van der Waals
69 (vdW) materials²¹⁻²³. Here we extend this to study excitonic absorption in the visible-near IR range.”

Comment 2:

In lines 174-189, the authors comment on the contributions of bright and dark excitons to the s-SNOM spectra. The authors note that excitons that are excited by light with polarization in the sample plane, are bright excitons and excitons that are excited with light polarization perpendicular to the sample plane, are dark excitons.

In particular the authors say “In contrast, the s-SNOM signal for atomically thin TMDs is governed by the in-plane dielectric function³⁰, which arises from bright excitons (see SOM).” This is misleading however. From their s-SNOM setup schematics in Fig. 1a it is clear that the incoming beam (at ~45 degrees) splits in to both an in-plane and out-of-plane components and both excite the sample, due to the antenna tip the out-of-plane component is actually the stronger of the two. Since the authors did not use any polarizer (either at the excitation or detection end) both excitations are active and signals are collected from the effect of both polarization axis. Also, the dielectric functions of the MLs in the out-of-plane and in-plane axis are not available to the authors, so they cannot know which axis of the sample they are exciting. See for example such axis dependent dielectric functions give different s-SNOM phase spectra, ACS Photonics 2021, 8, 1, 175–181. It seems from these experiments the authors cannot say why they could not see dark excitons (is it because they are weak or non-existent?) and why the bright excitons dominate (since the out-of-plane field is much stronger). For example would the authors have seen signature of dark exciton had they used a polarizer? The authors have to explain/modify their statements here or plan to use a polarizer, I believe the former solution is better.

Comment 3:

The authors point out in lines 231-240 that due to layer arrangements they see negligible hybridization between two atomic planes. This gives the impression that if the layer arrangements were right s-SNOM could pick up hybridization effect? If so, the authors have PFM capability (Pasupathy's group etc) and could have easily identified correct layer arrangements to see strong hybridization between two atomic planes? This reviewer is wondering then why such a sample was not chosen? If on the other hand, s-SNOM sensitivity was the reason then the authors should modify this statement to clarify for future readers.

Comment 4:

Is there any reason why the authors say-“our nano-IR” when what was done is visible and near-IR nanospectroscopy??

Comment 5:

In lines 262-273 the authors make several confusing remarks:

(i) somewhere up in the ms (lines 231-240) the authors argue that hybridization is negligible, now in lines 269-270 they say the reason for redshift is hybridization: “Therefore, the large shift observed here (21 meV) is likely to be caused by the interlayer hybridization.”

(ii) Somewhere up in the ms (lines 231-253) the authors argue on the importance of screening: “Therefore, we conclude that, in heterostructures with a large twist angle, the redshift of intralayer exciton resonance energies mainly arises from dielectric self screening^{9,47}”

and now in lines 271-273 they say:

“We note that the dielectric screening could also reduce the exciton resonance energy, but its effect is subtle (~10 meV, as demonstrated in last section), compared to the interlayer hybridization.”

The authors should carefully reconcile these statements and clearly describe why the effect of hybridization and screening are layer of arrangement dependence and make a clear consistent description. As it stands the arguments are confusing and not well supported by data.

Comment 6:

Also as a minor comment the authors say: In line 262 the authors say a one line statement of why there is a negligible oscillator strength, followed by Therefore...

I suggest to just remove line 262 and say:

The indirect transitions in

264 few-layer TMD crystals have negligible oscillator strength, and thus make only a minute
265 contribution to the dielectric function and/or to the s-SNOM signal. Therefore,.....

Reviewer #3 (Remarks to the Author):

In their manuscript Authors perform s-SNOM measurements on homo- and heterogeneous TMDs. By tuning the illumination from a visible TiSap laser they perform spectroscopic imaging and extract spectra of WSe₂ and MoSe₂ mono and multilayers, which they attribute to bright excitons and support this attribution by PL spectroscopy. Collected s-SNOM data is then utilized for the analysis of dielectric screening effects interlayer hybridization. It is supplemented by the theoretical analysis based on the dipole model, which is utilized to determine the dielectric function of the TMDs including damping constants.

The work is well elaborated, provides good discussion and detailed supplement materials. It is sufficiently novel for publishing in Nature Comms. and could attract broad interest.

I have only two concerns:

1) I am not convinced that the modeling of heterogeneous multilayer as a single layer with a single background contribution (epsilon_{bg}) is adequate. This concern is supported by different extracted values of the background permittivity depending on whether a mono/bi/trilayer was modeled ranging from 9.7 to 15.6 (epsilon_{bg} of Table 1). For example, why should the h-BN permittivity change because of WSe₂ or MoSe₂ monolayer? In my opinion, Authors should perform calculations with rp modelled as a multilayer and check if the extracted h-BN permittivity correspond to the literature value and that the corresponding exciton energy and damping are the same as derived by means of Eq. (1).

2) Authors write that the s-SNOM response is governed by the in-plane dielectric function. This is not fully clear to me. Due to radial component of the electric field, I would suspect that both in-plane and out-of-plane components of permittivity should show up in the s-SNOM response. This also follows from the discussion on page 3 of the supplement. So, what is the explanation for the strong claim in line 182-183? And why is TMV response should be from that of h-BN (at mid-IR), where both components are entering the s-SNOM response?

And one small comment:

3) The sentence in line 279 of the conclusion might be misinterpreted as that s-SNOM resolution is limited to above 20 nm. This is not so per multiple publications. It should be made clear that it is in this work the resolution was limited to 20 nm (due to the sharpness of utilized tip or the chosen step size). Otherwise, it could be misleading.

There also a small typo in line 268: “valance” -> “valence”.

In addition, formulas in the SOM should be proofread. I found numerous issues:

- i) σ -> σ_s in qq. below line 52
- ii) ϵ_j does not enter the equation for k_j^z in line 60. something is missing here.
- iii) equation in line 71 (ϵ_{ar} -> ϵ_a)
- iv) equation below line 78 (ϵ_{lison} -> ϵ_r)
- v) in line 85, I think the Authors mean dependence on k and not q , correct? Otherwise, please define k .

We thank reviewers for recognizing the novelty and importance of this work, which they describe as “novel and interesting” and “excellent” and deem “sufficiently novel for publishing in Nature Comms. and could attract broad interest”. We also appreciate the valuable and insightful comments and suggestion from reviewers, which further improve the quality of this work. In the revised version of the manuscript, we addressed each and every critical remark, as shown below.

Response to Reviewer #1:

The authors present experimental studies of s-SNOM imaging of excitons in monolayer MoSe₂, WSe₂, heterostructure and trilayer WSe₂ materials. They demonstrate measurements of dielectric functions and claim 20 nm spatial resolution. The results allowed for determining the 20 nm upper bound for dielectric screening length. The work is novel and interesting and may be published after the following comments are addressed:

We thank the reviewer for recognizing the novelty and interest of our work. We also appreciate the reviewer’s comments and suggestions. We addressed each critical remark in the revised version of manuscript.

1) On page 6 the authors claim that “data reveals that the prominent nano-optical contrast ... is rooted in the local excitonic response of TMD monolayers.” That is a reasonable expectation. However, is there an alternative explanation based on the tip shadowing effect? For example, the scattering signal at modulated harmonic is sensitive to the behavior of the tip. The tip oscillation changes across the junction between two materials because of different tip-sample interactions even in the absence of incident laser. Nonresonant scattered light will also give contrast. Resonant excitation may give larger contrast because of heating or charging effects. Both of these may give sub-diffraction contrast limited by the size of the tip even without localized excitation. So the claim of the local excitonic contribution is questionable. It would be helpful to clarify the mechanism behind the contrast. For example, the image showing lack of contrast for nonresonant excitation would prove the proposed local excitonic effect. For example, Figure 4a shows the presence of contrast between WSe₂ and hBN with nonresonant excitation. Does that contradict the proposed localized excitonic mechanism?

We thank the reviewer for carefully reading the manuscript and offering other possible explanations for the near-field signal contrast. The reviewer raised the possibility of shadowing effect and hypothesized that “scattering signal at modulated harmonic is sensitive to the behavior of the tip”. We were also concerned with a possibility of demodulation artifacts and for that reason we have demodulated the signal at the fourth and fifth harmonics. Earlier nano-optics experiments have established that even the third harmonic data provide genuine near-field data devoid of many artifacts (*Optics Communications* **182**, 321-328 (2000)). We confirmed that both the fourth and the fifth harmonic data exhibit nearly the same scattering contrast, as shown in Fig. R1. Therefore, we concluded that the contrast we observed does not depend on “modulated harmonic” and does not originate from tip shadowing effect. As we discussed in the main text and method section, demodulation at a high harmonic can better suppress the far-field background. Therefore, we focused on amplitude s_5 and phase φ_5 in the main text. In the revised supplementary materials, we added the mapping of amplitude s_4 and phase φ_4 in supplementary Fig. S5. In the revised version of the main text, we added references to the earlier works on higher harmonics demodulation (reference 26).

Fig. R1 The scattering amplitude and phase demodulated at fourth and fifth harmonics. a, b Representative near-field images of the normalized scattering amplitude $s_5(\omega)/s_5(h\text{-BN})$ and $s_4(\omega)/s_4(h\text{-BN})$, respectively. Boundaries of the monolayer WSe₂ region are displayed with black dashed lines. **c, d**, Near-field images of the normalized phase $\varphi_5(\omega) - \varphi_5(h\text{-BN})$ and $\varphi_4(\omega) - \varphi_4(h\text{-BN})$. The images with the same excitation energy are acquired simultaneously.

In addition, the reviewer also proposed that the heating effects or charge effects may give rise to contrast. The heating process from the continuous-wave laser is slow compared to the demodulation frequency. For the 5th harmonic demodulation, the demodulation frequency is around 400 kHz, which is much faster than the heating response. Regarding the charge effect, it cannot play a role in the near-field signal because in our charge neutral sample the photo-excited electron and hole densities are the same. More importantly, the heating and charge effect cannot explain the phase evolution of the scattered light. Therefore, the heating and charge effects can be safely ruled out. In the revised main text, we added the discussion addressing this important point (page 6).

We agree with the reviewer that the nonresonantly scattered light can give rise to near-field contrast between the hBN and TMDs, as shown in the Fig. 2 c and d. This contrast arises from the different dielectric function of hBN and TMDs. It should be noted that these contrasts at the

nonresonant energies do not vary as a function of photon energy, as shown in the Fig. R2. On the other hand, both the amplitude s_5 and phase φ_5 contrast prominently change around the excitation resonance energies. In short, the observed constant contrast away from exciton resonance energy does not contradict the proposed excitonic mechanism. Our data and the oscillator fit of the data confirm that the contrast arises from the dielectric function governed by the excitons.

Fig. R2 **a** and **b**, Normalized near-field amplitude $s_5(\omega)/s_5(h\text{-BN})$ and phase $\varphi_5(\omega) - \varphi_5(h\text{-BN})$ spectra for WSe_2 monolayer (data points). Fits to the spectra using the point dipole model (PDM) and a single Lorentz oscillator (Eq. 1 in main text) are shown as solid curves. The near-field contrast between the h-BN and WSe_2 monolayer is almost constant when the photon energy is far lower than exciton resonance energy.

In the revised manuscript, we followed reviewer’s suggestions by further clarifying the mechanism of the s-SNOM scattering contrast. The key points are summarized here. The s-SNOM scattering signal is determined by the reflectance of p-polarized light (r_p) from the sample, that is, $s \approx \int W(k)r_p dk$, where $W(k)$ is the weighting function and determined by the tip geometry (*Nano Letters* **11**, 4701-4705 (2011), *Nature Materials* **14**, 421-425 (2015)). For the atomically thin materials on thick substrates, r_p is determined solely by the in-plane dielectric function (*Physical Review B* **90**, 205422 (2014)). Therefore, the s-SNOM scattering signal is governed by the in-plane dielectric function.

2) Why do the authors use large scale averaging to obtain dielectric function spectra? Is it because of low SNR? Does that limit the application of s-SNOM?

We thank the reviewer for pointing out the large scale averaging we used and his/her concern about the impact of the signal to noise ratio on s-SNOM applications. As shown in the s-SNOM images, the contrast between various regions/terraces can be clearly identified even without spatial averaging. Notably, the contrast between distinct regions exceeds relatively minor variations of the signal within any given region (Figs. 2a, b). This latter aspect of the data justifies averaging when analyzing the spectral characteristics of each region. The reviewer is correct: averaging does improve the signal noise ratio of the s-SNOM spectra and thus averaging facilitates a faithful determination of the dielectric function. We do not believe that our analysis limits in any way “the application of s-SNOM”. S-SNOM images can clearly resolve all nanometer-wide boundaries between the terraces and some terraces appear to be in the deeply sub-diffractive regime. To address this important point, we expanded the discussion in the revised main text (page 7).

3) The authors claim uniform distribution of s-SNOM signal over the flakes. That contradicts the previous findings of TEPL that showed heterogeneous local excitonic response. How do the authors interpret it? In fact, from the s-SNOM images it seems that there are some heterogeneous structures on top of the uniform background. They might even be correlated for different excitations. Did the authors analyze such correlations?

We thank the reviewer for bringing up tip-enhanced photoluminescence (TEPL) results. As discussed in the introduction section of the main text, TEPL and the s-SNOM are two complementary approaches to investigate excitons. Whereas the s-SNOM spectra are directly determined by the dielectric function of the materials; TEPL is determined by both the exciton population and the transition rate and therefore is sensitive to sample parameters (*Nature Physics* **15**, 1140-1144 (2019)). Stain, defects and contaminants in samples, particularly CVD grown ones commonly used in TEPL experiments, can impact the exciton distribution and resonance energies and thus give rise to inhomogeneity of the TEPL. In our work, to probe the intrinsic properties of TMDs, the samples were exfoliated from the high quality crystals grown by the self-flux method (*Nano Lett.*, **19**, 4371-4379 (2019)) and fabricated by the contamination-free stacking methods (details in Method). The resulting high-quality sample possesses nearly uniform dielectric function, thereby exhibiting the nearly homogenous s-SNOM signal. As the reviewer pointed out, “it seems that there are some heterogeneous structures”. But these features are too faint and do not allow us to perform statistically significant analysis. In response to comment #1.3, in the revised main text (page 10), we added a paragraph to address this important point.

4) In Table 1 what is the meaning of ϵ_{bg} and why is it different for different materials? Is it related to the dielectric function of hBN that should have the same value for any location?

We appreciate the reviewer for pointing out the possible confusing definition of ϵ_{bg} . In our work, the ϵ_{bg} denotes the background dielectric function of the TMDs and it originates from the virtual high energy transitions (Epstein, I. *et al. 2D Materials* **7**, 035031 (2020); Scuri, G. *et al. Physical Review Letters* **120**, 037402 (2018)). Therefore, this ϵ_{bg} is a material dependent parameter. This parameter is not related to the dielectric function of the substrate hBN. In order to eliminate possible confusion we replaced ϵ_{bg} with ϵ_{∞} . In addition, the definition and the origin of ϵ_{∞} were clarified after Eq. (1) of the main text (page 8).

5) The claims of nanoscale resolution are questionable in the context of resonance energies and dielectric functions. There is no data in the manuscript that shows either of them with nanoscale resolution. The only data are for the s-SNOM signals at one excitation energy at a time. The authors should either revise the claims in the abstract and summary or add more data. For example, Figure 3e shows a promising result but in microscale. Did the authors try to zoom in or the SNR is too small?

We thank reviewer for his/her concerns on the resolution. Our data reveal abrupt boundaries between distinct regions in our samples. This contrast is formed by the variation of the dielectric function. Figs.4 a,b display representative results affirming that the contrast is changing on the length scale about 20 nm. Data in Fig.2a,b uncover the abruptness of these changes both in amplitude and phase data over the entire frequency range. The totality of data in Fig.2-4 attest to

the nano-scale resolution of dielectric function. We added a sentence of page 13 of the revised main text to emphasize these aspects of the data.

Response to Reviewer #2 (Remarks to the Author):

- What are the noteworthy results? This is an excellent work where the authors use s-SNOM technique coupled with near-infrared and visible frequency lasers to perform spectroscopy and map the real and imaginary parts of the dielectric functions of ML MoSe₂, WSe₂, and their heterostructures. They implement the method of the equivalence of far-field absorption spectra with the s-SNOM phase spectra to study excitons in these structures. Along the way they also found the effect of dielectric screening on the dielectric function.

- Will the work be of significance to the field and related fields? How does it compare to the established literature? If the work is not original, please provide relevant references.

The work is significant in the sense that nano-spectroscopy in MLs and heterostructures has not been applied before and this now opens a way to study excitons at high-resolution. • Does the work support the conclusions and claims, or is additional evidence needed?

The work is supported by the conclusions and claims, the reviewer has made some comments to improve the presentation of the work.

- Are there any flaws in the data analysis, interpretation and conclusions? Do these prohibit publication or require revision?

No

- Is the methodology sound? Does the work meet the expected standards in your field?

Yes

- Is there enough detail provided in the methods for the work to be reproduced?

Yes

We thank the reviewer for the acknowledgement of the importance of this work and providing his/her valuable comments and suggestions.

Comment 1:

In lines 68-70 the authors say:

“This latter capability has been extensively used to study plasmons and phonons in other classes of van der Waals (vdW) materials. However, formidable technical challenges so far precluded quantitative nano-spectroscopy of excitonic absorption by means of s-SNOM.”

I believe what the authors mean by formidable technical challenge is just simply lack of stable lasers in the visible and near-IR frequencies, nothing else. Also note similar visible/near-IR nano-spectroscopy with s-SNOM has been done before, although not on MLs see for example APPLIED PHYSICS LETTERS 96, 063107 (2010). For all these reasons I would simply say: “This latter capability has been extensively used to study plasmons and phonons in other classes of van der Waals (vdW) materials²¹⁻²³. Here we extend this to study excitonic absorption in the visible-near IR range.”

We thank the reviewer for pointing out this valuable reference. Regarding “the formidable technical challenges”, as the reviewer stated, the laser stability is indeed a challenge for nanospectroscopy of excitons. However, to realize the nano-spectroscopy of excitons in atomically thin TMDs, there are other technical challenges to overcome. For example, when the wavelength of the incident light decreases from ~10 μm (MIR) to less than 1 μm (Vis), because of the resulting reduced focus diameter, more scattered light will come from the cantilever or shank of the AFM tip, thus making it more challenging to extract the genuine near-field signal. In addition, the s-SNOM phase is more prone to drift of the optical path length when the wavelength is reduced.

The reference pointed out by the reviewer (*Applied Physics Letters* **96**, 063107 (2010)) is an important step towards quantitative nanospectroscopy of excitons. In the revised manuscript, we added a reference to this article (Ref. 24) and expanded our discussions of experimental challenges of the nanospectroscopy of excitons (page 4).

Comment 2:

In lines 174-189, the authors comment on the contributions of bright and dark excitons to the s-SNOM spectra. The authors note that excitons that are excited by light with polarization in the sample plane, are bright excitons and excitons that are excited with light polarization perpendicular to the sample plane, are dark excitons.

In particular the authors say “In contrast, the s-SNOM signal for atomically thin TMDs is governed by the in-plane dielectric function³⁰, which arises from bright excitons (see SOM).” This is misleading however. From their s-SNOM setup schematics in Fig. 1a it is clear that the incoming beam (at ~45 degrees) splits in to both an in-plane and out-of-plane components and both excite the sample, due to the antenna tip the out-of-plane component is actually the stronger of the two. Since the authors did not use any polarizer (either at the excitation or detection end) both excitations are active and signals are collected from the effect of both polarization axis. Also, the dielectric functions of the MLs in the out-of-plane and in-plane axis are not available to the authors, so they cannot know which axis of the sample they are exciting. See for example such axis dependent dielectric functions give different s-SNOM phase spectra, *ACS Photonics* 2021, 8, 1, 175–181. It seems from these experiments the authors cannot say why they could not see dark excitons (is it because they are weak or none-existent?) and why the bright excitons dominate (since the out-of-plane field is much stronger). For example would the authors have seen signature of dark exciton had they used a polarizer? The authors have to explain/modify their statements here or plan to use a polarizer, I believe the former solution is better.

We thank reviewer for pointing out the issue related to the contributions of the in-plane and out-of-plane dielectric functions to the s-SNOM signal. In the revised manuscript, we followed reviewer’s suggestions by further clarifying the mechanism of the s-SNOM scattering contrast. Key points are summarized here.

A. On the in-plane and out-of-plane component of the electric field. In the s-SNOM experiment, only the electric field along the tip shank direction dominates the scattering signal. It is because tip polarization along its shank direction is much larger than that along its perpendicular direction. In addition, for the electrical field perpendicular to the tip (parallel to the sample), the tip polarization and its image dipole are anti-parallel and thus nearly cancel the total scattering (*Optics*

Communications **182**, 321-328 (2000)). Moreover, in our experiments, the asymmetric Michelson interferometry was applied to measure the near-field signal. The signal registered by detectors is the interfering light between p-polarized light from reference arm (E_r) of the Michelson interferometer and the scattered light by tip (E_t), this is, $s \propto E_r E_t$. As only light with same polarization can interfere, the vertically polarized light from the reference arm ensures that the only p-polarized scattered light can be detected.

B. On the in-plane and out-of-plane dielectric function. The s-SNOM scattering signal is determined by the reflectance of p-polarized light (r_p) from the sample, that is, $s \approx \int W(q)r_p dq$, where $W(q)$ is the weighting function and determined by the tip geometry (*Nano Letters* **11**, 4701-4705 (2011), *Nature Materials* **14**, 421-425 (2015)). For the atomically thin material on the thick substrate, r_p is determined by the in-plane dielectric function (*Physical Review B* **90**, 205422 (2014); *2D Mater.* **5**, 025021 (2018)). Therefore, the s-SNOM scattering signal is governed by the in-plane dielectric function.

C. On the bright and dark excitons. As bright excitons can be excited by light with in-plane polarization, these bright excitons govern the in-plane dielectric function of the TMDs. For that reason, the bright excitons define the behavior of the r_p , thereby governing the s-SNOM scattering signal. Thus, the dark excitons cannot contribute significantly to the s-SNOM scattering signal.

Yet, for thicker sample, both the in-plane and out-of-plane dielectric function can contribute to the s-SNOM signal, as in the work pointed out by reviewer (*ACS Photonics* **8**, 1, 175–181 (2021)). The dielectric function curve reproduced from this reference, reveals both in-plane and an out-of-plane phonon modes in mica (in Fig. R3 a and b). However, as shown in Fig. R3 c and d, for the samples thinner than 3 nm, the s-SNOM signal originating from the out-of-plane phonon mode around 900 cm^{-1} cannot be discriminated. In summary, the results on mica are consistent with our conclusions for monolayer thick TMDs: in atomically thin TMDs the s-SNOM signal is governed by the in-plane dielectric function. In the revised version, we added a reference of this work to this article (Ref. 44) and expand the discussion on contributions of out-of-plane dielectric function to the s-SNOM signal.

Fig. R3 s-SNOM response of in-plane and out-of-plane dielectric functions in muscovite mica. **a**, **b** Real and imaginary parts of the permittivity of the in-plane and out-of-plane components. **c**, **d** Nano-FTIR amplitude and phase spectra of mica on SiO₂ substrate at different thickness. All panels are adapted from ACS Photonics **8**, 175–181 (2021).

Comment 3:

The authors point out in lines 231-240 that due to layer arrangements they see negligible hybridization between two atomic planes. This gives the impression that if the layer arrangements were right s-SNOM could pick up hybridization effect? If so, the authors have PFM capability (Pasupathy's group etc) and could have easily identified correct layer arrangements to see strong hybridization between two atomic planes? This reviewer is wondering then why such a sample was not chosen? If on the other hand, s-SNOM sensitivity was the reason then the authors should modify this statement to clarify for future readers.

We thank reviewer for his/her interest in the s-SNOM's capability of probing the hybridization in TMDs. The hybridization can indeed be detected by the s-SNOM spectra if the two layers are well aligned. Our results for the trilayer WSe₂, in which the adjacent layers are well aligned, uncover nano-optical signatures of hybridization. The reviewer is absolutely correct that PFM can well investigate the moiré patterns that occur in twisted vdW materials. The goals of the present work were to firmly establish and quantify the ability of S-SNOM to reveal the excitonic response of monolayer TMDs. Research on twisted TMDs is outside of the scope of this manuscript but will be carried out in follow up experiments, as we point out on page 14 of the revised main text.

Comment 4:

Is there any reason why the authors say-"our nano-IR" when what was done is visible and near-IR nanospectroscopy?

We thank the reviewer for pointing out this issue. We changed "nano-IR" to "nano-NIR/Vis" in the revised version.

Comment 5:

In lines 262-273 the authors make several confusing remarks:

(i) somewhere up in the ms (lines 231-240) the authors argue that hybridization is negligible, now in in lines 269-270 they say the reason for redshift is hybridization: "Therefore, the large shift observed here (21 meV) is likely to be caused by the interlayer hybridization."

(ii) Somewhere up in the ms (lines 231-253) the authors argue on the importance of screening: "Therefore, we conclude that, in heterostructures with a large twist angle, the redshift of intralayer exciton resonance energies mainly arises from dielectric self screening^{9,47}"

and now in lines 271-273 they say: "We note that the dielectric screening could also reduce the exciton resonance energy, but its effect is subtle (~10 meV, as demonstrated in last section), compared to the interlayer hybridization."

The authors should carefully reconcile these statements and clearly describe why the effect of hybridization and screening are layer of arrangement dependence and make a clear consistent description. As it stands the arguments are confusing and not well supported by data.

We thank the reviewer for pointing out the confusion. We believe the confusion comes from the discussion on two different materials. In lines 231-253, we discussed excitons in misaligned WSe₂/MoSe₂ heterostructure, while in lines 254-273, we focused on another topic—excitons in 2H-stacked WSe₂ trilayer. For WSe₂/MoSe₂ heterostructure and WSe₂ trilayer, the screening and hybridization play different roles. That is why in lines 231-253, we emphasized the screening, while in 254-273 we highlighted the hybridization.

(i) In line 231-253, we discussed the exciton in misaligned MoSe₂/WSe₂ heterostructure. In this case, the exciton red-shift is dominated by the screening. Therefore, the conclusion is that “hybridization is negligible”. Then, in lines 254-273, we discussed the exciton in WSe₂ trilayer. In this 2H stacking WSe₂ trilayer, the interlayer hybridization plays an important role in the exciton resonance energy red-shift. “Therefore, the large shift observed here (21 meV) is likely to be caused by the interlayer hybridization.”

(ii) In lines 231-253, we discussed the exciton of constituent layers in misaligned MoSe₂/WSe₂ heterostructure and argued on the importance of screening; while in line 271-272, the topic shifted to WSe₂ trilayer. In WSe₂ trilayer, “We note that the dielectric screening could also reduce the exciton resonance energy, but its effect is subtle (~10 meV, as demonstrated in last section), compared to the interlayer hybridization.”

In response to comment #2.5, in the revised manuscript we modified the discussion to avoid confusion/misinterpretation of the statements.

Comment 6:

Also as a minor comment the authors say: In line 262 the authors say a one line statement of why there is a negligible oscillator strength, followed by Therefore...

I suggest to just remove line 262 and say:

The indirect transitions in
264 few-layer TMD crystals have negligible oscillator strength, and thus make only a minute
265 contribution to the dielectric function and/or to the s-SNOM signal. Therefore,.....

We thank the reviewer for making the statement more concise. We respectfully took the reviewer’s suggestion in the revised version by removing line 262.

Response to Reviewer #3 (Remarks to the Author):

In their manuscript Authors perform s-SNOM measurements on homo- and heterogeneous TMDs. By tuning the illumination from a visible Ti:Sap laser they perform spectroscopic imaging and extract spectra of WSe₂ and MoSe₂ mono and multilayers, which they attribute to bright excitons and support this attribution by PL spectroscopy. Collected s-SNOM data is then utilized for the analysis of dielectric screening effects interlayer hybridization. It is supplemented by the theoretical analysis based on the dipole model, which is utilized to determine the dielectric function of the TMDs including damping constants. The work is well elaborated, provides good discussion and detailed supplement materials. It is sufficiently novel for publishing in Nature Comms. and could attract broad interest.

We thank the reviewer for the positive remarks and for recommending publication of our work. The reviewer’s comments and suggestions are appreciated.

I have only two concerns:

1) I am not convinced that the modeling of heterogenous multilayer as a single layer with a single background contribution (epsilon_bg) is adequate. This concern is supported by different extracted values of the background permittivity depending on whether a mono/bi/trilayer was modeled ranging from 9.7 to 15.6 (epsilon_bg of Table 1). For example, why should the h-BN permittivity change because of WSe2 or MoSe2 monolayer? In my opinion, Authors should perform calculations with rp modelled as a multilayer and check if the extracted h-BN permittivity correspond to the literature value and that the corresponding exciton energy and damping are the same as derived by means of Eq. (1).

We thank the reviewer for pointing out his/her concern about the model. Consistent with reviewer's suggestion, the multilayer model was indeed used to calculate the r_p with transfer matrix method. We believe that the main confusion comes from the definition of ϵ_{bg} . The ϵ_{bg} in the manuscript denotes the background dielectric function of the TMDs, rather than the dielectric function contributions from the substrate. This TMD background dielectric function ϵ_{bg} originates from the virtual high energy transitions (Epstein, I. *et al. 2D Materials* **7**, 035031 (2020); Scuri, G. *et al. Physical Review Letters* **120**, 037402 (2018)). Therefore, this ϵ_{bg} is a material dependent parameter. This is why the extracted ϵ_{bg} values depend on the TMD materials (Table 1 in the main text). To eliminate confusion, we replaced ϵ_{bg} with ϵ_∞ . In addition, the definition and the origin of ϵ_∞ were clarified after Eq. (1) of the main text (page 8).

2) Authors write that the s-SNOM response is governed by the in-plane dielectric function. This is not fully clear to me. Due to radial component of the electric field, I would suspect that both in-plane and out-of-plane components of permittivity should show up in the s-SNOM response. This also follows from the discussion on page 3 of the supplement. So, what is the explanation for the strong claim in line 182-183? And why is TMV response should be from that of h-BN (at mid-IR), where both components are entering the s-SNOM response?

We thank the reviewer for this remark. The reviewer's concern provided us with an opportunity to clearly address the issue of in-plane and out-of-plane response in the revised version. The conclusion is that the s-SNOM response of atomically thin materials is governed by the in-plane dielectric function.

The s-SNOM scattering signal is determined by the reflectance of p-polarized light (r_p) from the sample, that is, $s \approx \int W(q)r_p dq$, where $W(q)$ is the weighting function and determined by the tip geometry (*Optics Communications* **182**, 321-328 (2000), *Nano Letters* **11**, 4701-4705 (2011), *Nature Materials* **14**, 421-425 (2015)). For the atomically thin material on a thick substrate, this thin material can be treated as a conductive sheet with negligible thickness when one calculates r_p using Maxwell boundary conditions. Thus, the r_p is only determined by the in-plane dielectric function (*Physical Review B* **90**, 205422 (2014); *2D Mater.* **5**, 025021 (2018)). As a consequence, the s-SNOM scattering signal is solely governed by the in-plane dielectric function. In contrast, the out-of-plane dielectric function of the atomically thin TMDs plays a negligible role in r_p . Thus, this out-of-plane dielectric function does not contribute the s-SNOM scattering signal. This discussion was included in the supplementary materials.

When the material thickness increases, the materials cannot be simply treated as a conductive sheet when one calculates r_p . Both the in-plane and the out-of-plane dielectrics contribute to the s-SNOM response. However, we note that the contribution from the out-of-plane response has a small weight, about d/λ smaller than that from in-plane, where d is the layer thickness and λ is the typical in-plane length scale of the tip field.

And one small comment:

3) The sentence in line 279 of the conclusion might be misinterpreted as that s-SNOM resolution is limited to above 20 nm. This is not so per multiple publications. It should be made clear that it is in this work the resolution was limited to 20 nm (due to the sharpness of utilized tip or the chosen step size). Otherwise, it could be misleading.

We thank the reviewer for pointing out the possible misleading statement on s-SNOM resolution. We totally agree with the reviewer that the resolution of s-SNOM can be better than 20 nm. The 20 nm is the achieved resolution in our work. In our response to reviewer's comment #3.1, we respectfully took the reviewer's suggestion and revised the statement on s-SNOM resolution. Now the revised statement is that "We obtained the upper bound of the dielectric screening length of 20 nm limited by the achieved spatial resolution of s-SNOM in this work."

There also a small typo in line 268: "valance" -> "valence".

In addition, formulas in the SOM should be proofread. I found numerous issues:

i) $\sigma \rightarrow \sigma_s$ in qq. below line 52

ii) ϵ_j does not enter the equation for k_j^z in line 60. something is missing here.

iii) equation in line 71 ($\epsilon_{ar} \rightarrow \epsilon_a$)

iv) equation below line 78 ($\epsilon_{lison} \rightarrow \epsilon_r$)

v) in line 85, I think the Authors mean dependence on k and not q , correct? Otherwise, please define k .

We thank the reviewer for carefully reading the manuscript and pointing out the errors and typos. We have revised them accordingly and thoroughly proofread the manuscript. Now we used the subscript "r" to denote the relative permittivity.

REVIEWER COMMENTS

Reviewer #1 (Remarks to the Author):

The authors satisfactorily responded to my comments.

Reviewer #2 (Remarks to the Author):

all comments are addressed fully. I have no more comments. The paper can be published from my side as is.

Reviewer #3 (Remarks to the Author):

Thank you for your insightful responses. I have no objections and recommend accepting the manuscript for publication as is.